# Government Options to Reduce the Impact of Alcohol on Human Health: Obstacles to Effective Policy Implementation

**DOI:** 10.3390/nu13082846

**Published:** 2021-08-19

**Authors:** Tim Stockwell, Norman Giesbrecht, Kate Vallance, Ashley Wettlaufer

**Affiliations:** 1Canadian Institute for Substance Use Research, University of Victoria, Victoria, BC V8P 5C2, Canada; vallance@uvic.ca; 2Centre for Addiction and Mental Health, Toronto, ON M5V 2B4, Canada; Norman.Giesbrecht@camh.ca (N.G.); Ashley.Wettlaufer@camh.ca (A.W.)

**Keywords:** alcohol, policy, public health, alcohol industry, Canada

## Abstract

Evidence for effective government policies to reduce exposure to alcohol’s carcinogenic and hepatoxic effects has strengthened in recent decades. Policies with the strongest evidence involve reducing the affordability, availability and cultural acceptability of alcohol. However, policies that reduce population consumption compete with powerful commercial vested interests. This paper draws on the Canadian Alcohol Policy Evaluation (CAPE), a formal assessment of effective government action on alcohol across Canadian jurisdictions. It also draws on alcohol policy case studies elsewhere involving attempts to introduce minimum unit pricing and cancer warning labels on alcohol containers. Canadian governments collectively received a failing grade (F) for alcohol policy implementation during the most recent CAPE assessment in 2017. However, had the best practices observed in any one jurisdiction been implemented consistently, Canada would have received an A grade. Resistance to effective alcohol policies is due to (1) lack of public awareness of both need and effectiveness, (2) a lack of government regulatory mechanisms to implement effective policies, (3) alcohol industry lobbying, and (4) a failure from the public health community to promote specific and feasible actions as opposed to general principles, e.g., ‘increased prices’ or ‘reduced affordability’. There is enormous untapped potential in most countries for the implementation of proven strategies to reduce alcohol-related harm. While alcohol policies have weakened in many countries during the COVID-19 pandemic, societies may now also be more accepting of public health-inspired policies with proven effectiveness and potential economic benefits.

## 1. Introduction

Alcohol and public health policy as an organised field of academic inquiry is relatively new. In 1975, the assertion by Kettil Bruun, Griffith Edwards, Robin Room and others that alcohol was a public health issue was seen as new and controversial [1]. Using case studies mostly from Scandinavia and other parts of Europe, they provided evidence that the total consumption of alcohol was a reliable predictor of the extent of serious alcohol-related harms in any particular population, e.g., liver disease, injury and alcohol use disorders. Perhaps their most revolutionary proposal was that governments could take action to reduce alcohol-related harm, principally by reducing its affordability (e.g., through higher taxes) and its availability (e.g., by reduced days and hours of sale).

In the decades since this landmark publication, often referred to as the “Purple Book”, thousands of studies have been published on the intersections between population consumption of alcohol, the extent of population harm from alcohol and the relative effectiveness of alternative government policies. Learnings from these research mountains are now distilled in systematic reviews and meta-analyses and are further informed by advances in theory and methodology. For example, the International Model of Alcohol Harms and Policies (InterMAHP) [2] provides an online tool to accurately estimate the extent of alcohol-related harm in any population (be it a city, region, or country) from available data on per capita alcohol consumption and the recorded prevalence of various causes of death, injury, and illness. InterMAHP also helps to estimate how these harms change when policies are introduced to increase or decrease alcohol consumption (e.g., [3]). The tool is built on some solid foundations including (a) a demonstrated mathematical relationship between the average consumption of a population and the distribution of drinkers according to their typical daily consumption [4] and (b) the latest systematic reviews and meta-analyses of published studies estimating risks of different types of disease and injury as a function of how much people drink.

As a result of such advances, it is now possible to provide very specific and detailed advice to governments regarding the public health consequences of policy decisions in such concrete terms as how many people will become ill, injured or die prematurely from alcohol-related reasons if policy X or Y is introduced—and what might be the economic costs and benefits. Furthermore, the Sheffield Alcohol Policy Model (SAPM, e.g., [5]) has been effectively applied over the last decade to inform legal and policy processes culminating in the introduction of a Minimum Unit Price (MUP) for alcohol in Scotland in May 2018 [6], a successful policy which is rapidly being emulated in other countries (e.g., [7,8,9]). Both InterMAHP and SAPM have been used to estimate policy impacts of MUP on such sensitive outcomes as government revenues and consumer expenditure (e.g., [3,10]).

In this paper, we take the view that while there are considerable obstacles and challenges to the implementation of effective alcohol policies, theories and methodologies to inform effective implementation are available to policy decision makers, the public health field and the broader community of concerned citizens. While authors of the most highly cited and comprehensive evidence review on effective alcohol policies [11] concluded, rather depressingly, that effective policy is inevitably unpopular, we take a more optimistic approach. We highlight “circuit breaker” strategies which can create a more favourable climate for governments being prepared to implement policies with demonstrated effectiveness in reducing alcohol-related harms.

## 2. What Is Effective Alcohol Policy?

The World Health Organization (WHO) recently launched the WHO-SAFER international initiative to reduce alcohol-related harm in member countries [12]. Built around a growing evidence base and the pre-existing WHO Global Strategy on Alcohol [13], this initiative promotes the following objectives:(1)Strengthening restrictions on alcohol availability(2)Advancing and enforcing drink-driving countermeasures(3)Facilitating access to screening, brief interventions and treatment(4)Enforcing bans or comprehensive restrictions on alcohol advertising, sponsorship and promotion(5)Raising prices on alcohol through excise taxes and pricing policies.

A number of research-led initiatives have sought to conduct systematic assessments of the extent to which such evidence-based strategies are being implemented in jurisdictions and, in some cases, relate the extent of policy implementation to outcomes (e.g., [14,15]). The co-authors of this manuscript developed the Canadian Alcohol Policy Evaluation (CAPE) project [16,17,18,19] in which 250 indicators of alcohol policy implementation were evaluated across 11 domains of effective government action in each of Canada’s 10 provinces and three territories. Table 1 below identifies seven domains for which there is evidence of direct effectiveness for the reduction in alcohol consumption and harms and also indicates the strength of this evidence and the likely scope of impacts on population level harms. These domains include each of the five WHO SAFER domains listed above and, in addition, two others with long-standing evidence to support them: Liquor Law Enforcement (e.g., targeting enforcement to high-risk premises and enforcing laws prohibiting service to intoxicated persons) and Minimum Legal Drinking Age Laws.

Uniquely, CAPE also identified and assessed four “indirect policy domains” with the capacity to facilitate the implementation of the other seven directly effective policies. In short, the extent of government ownership of alcohol distribution and retail systems (i.e., Alcohol Control System) facilitates control of direct policy levers such as pricing and availability; a comprehensive Alcohol Strategy outlining multiple policy objectives with funding for implementation, clear leadership independent of commercial vested interests and an evaluation plan will guide multiple arms of government towards effective action; comprehensive, transparent and regular public Monitoring and Reporting of alcohol harms and policies will help to sustain and continue to direct effective policies; Health and Safety Messaging, particularly at the point of purchase and consumption through container labelling, will motivate effective policy action by highlighting risks and harms from alcohol consumption, both among citizens and their elected representatives [17].

## 3. What Are the Obstacles to Effective Policy Implementation?

There are some recurring themes that characterise resistance to the implementation of the types of effective alcohol policies discussed above. We will restrict our discussion to four key themes: (1) a lack of awareness about the extent of alcohol-related harm and the effectiveness of alcohol policies; (2) a lack of government regulatory and legislative structures focused on reducing harm from alcohol; (3) effective lobbying by alcohol industry groups to foster such skepticism and to propose less-effective policies; (4) absent or ineffective lobbying by public health advocacy groups. We will discuss each of these in turn and suggest ways such impediments can be overcome or at least be rendered less obstructive.

### 3.1. Low Public Awareness of the Extent of Alcohol Related Harm

Alcohol use is causally implicated in several hundred diagnostic categories of illness and injury [2] including cancer and cardiovascular diseases, the two most common causes of death in developed countries. However, alcohol’s contribution to the illnesses and injuries leading to emergency department presentation, hospitalisation and/or premature death are usually not recorded on death certificates or diagnostic records. Special studies are required to estimate the extent to which these premature deaths and presentations might be attributable to alcohol. The methods of such studies are complex and technical, well beyond the grasp of many health professionals, let alone decision-makers or members of the public. There is also a widespread belief evidenced by national surveys and perpetuated by industry groups that moderate alcohol consumption can protect people from a wide array of harms, cardiovascular diseases in particular. For example, one survey found that 57% of the Canadian population believed alcohol in moderation was good for their health (e.g., [20]). The prevalence of this belief may have waned in recent years, perhaps reflecting more open debate and critique of this idea, with a 2020 survey of Quebecers finding this proportion to now be lower, at 40% [21]. Opinion surveys in different countries have also frequently found that when people are asked to estimate whether alcohol or illicit drugs contribute the most problems to society, usually alcohol comes in a poor second [22]. 

Contrary to these popular and governmental biases, the hard evidence is that alcohol use contributes more health harms than other psychoactive drugs that are prohibited in many countries (i.e., cannabis, opioids, stimulants, etc.). This is borne out of WHO Global Burden of Disease estimates which most recently indicate that 2.4 million premature deaths globally are attributable to alcohol, compared with 0.49 million for illicit drugs in 2019 [23]. In Canada, the ongoing Canadian Substance Use Costs and Harms [24] project reports annually on estimated hospitalisations, deaths and economic costs from alcohol, cannabis, nicotine, opioids and other psychoactive substances [24]. The overall economic cost of substance use across healthcare, lost productivity, criminal justice and other direct-cost domains was estimated at $46 billion in 2017. Accounting for more than three-quarters of the total cost are Canada’s three legal drugs: alcohol ($16.6 billion), tobacco ($12.3 billion) and cannabis ($3.2 billion). Among illicit substances, only opioids ($5.9 billion) and cocaine ($3.7 billion) make up more than 5% of the total cost [24]. This disparity persists even at a time when there is an opioid overdose crisis. Typically, public debate focuses on the “crises” even if the harm and death rates are not substantially higher than those from alcohol.

Part of the reason for the lack of awareness of alcohol-related harms may be the extent to which these are not wholly attributable to the use of alcohol. Alcohol’s contributions to serious illnesses, injuries and premature deaths are usually contributory and often not even recorded. Sherk et al. [25] estimated that over 90% of alcohol-attributable deaths in a Canadian jurisdiction were partially alcohol attributable, and less than 10% wholly or 100% attributable. A good example is alcohol-related breast cancer. It has been estimated that approximately 7% of these cancers can be attributed to alcohol [26]. Nonetheless, despite this relatively small contribution, because of the high prevalence of cancer in general, about 30% of alcohol-attributable deaths in Canada are from alcohol-related cancers [24] (Note that these estimates sum up the fractions of lives lost or hospitalisations caused to make up each partially attributable case). 

Not surprisingly, there is evidence worldwide that few members of the public are aware that alcohol is a carcinogen, despite the WHO’s International Agency for Research on Cancer having confirmed a causal association more than 30 years ago [27]. For example, a national UK survey found that only 13% of respondents could identify alcohol as a cause of cancer without prompting, 33% could do so if prompted and 54% were completely unaware of any alcohol–cancer connection [28].

The lack of public awareness of the hidden, partially alcohol-attributable illnesses and injuries is also reflected in the widespread perception that the only serious harms from alcohol are alcohol dependence, liver cirrhosis and crashes from impaired driving. In fact, these are just tips of much larger icebergs. At the outset of the COVID-19 pandemic, there were several North American examples of perverse rationales being given by governments for classifying alcohol as an “essential commodity”, which reflect this restricted view of alcohol-related harm. In California [29] and at least two Canadian jurisdictions [30], this essential-commodity status was justified as a way of preventing healthcare services from being overwhelmed by people going into alcohol withdrawal. In Canada, at most only 5% of alcohol-attributable hospitalisations are associated with alcohol dependence or withdrawal [30]. When India imposed a strict lockdown with a complete alcohol prohibition, there was indeed observed a spike in presentations for alcohol withdrawal, but this was transient and soon demand dropped to zero [31]. It was also documented during a Nordic strike of government alcohol monopoly workers that demand for alcohol withdrawal treatment dropped to less than 40% of normal levels [32]. It has been shown elsewhere [30] that when partial as well as wholly alcohol-attributable hospitalisations are considered, maintaining alcohol use imposes a greater burden on healthcare services in many countries than has COVID-19, e.g., 105,000 alcohol-attributable hospital admissions were estimated for 2017 [24] compared with 40,000 COVID-19-related admissions in the first full year of the pandemic [33]. 

### 3.2. Low Public Awareness of the Effectiveness of Alcohol Policies

Babor et al. [11] advanced the influential narrative that effective alcohol policies (principally restrictions on pricing and availability) are unpopular with the public and their elected representatives, while less effective educational strategies receive the strongest support in opinion surveys. This position is well based on opinion research spanning multiple countries which consistently suggests that across-the-board price increases in particular are the most unpopular of all alcohol policy options. For example, a careful analysis of public opinion on alcohol policy in the UK found 84% support for public information campaigns, whereas only 33% supported increasing the price of alcohol, 41% for reducing the number of liquor outlets and 39% for reducing their hours of trading [34]. Critically, Li et al. [34] were able to identify skepticism about the effectiveness of different policies as a determining factor in whether they would be supported. 

There is also a widespread belief that “alcoholics” will always access alcohol regardless of price or availability, a viewpoint that anyone who has discussed alcohol control policies with decision-makers or members of the public will have encountered. This is contrary to evidence from a thorough systematic review that heavy drinkers do reduce their drinking when prices increase [35] and from the accounts of alcohol-dependent drinkers themselves [36].

### 3.3. Absent or Inadequate Government Regulatory and Legislative Structures

Governments in both the developed and developing worlds rarely give high level consideration to alcohol policy. Responsibility for the implementation of the kinds of evidence-based policies listed above is usually scattered across multiple ministries and departments. Health ministries are mostly responsible for providing healthcare to the sick and injured rather than addressing underlying causes and reducing risk factors, thus they do not have direct access to the most important policy levers, namely, price, availability and marketing restrictions. Finance departments are responsible for taxation and pricing but are not mandated to reduce adverse health and safety impacts of alcohol and often are unaware of the economic costs of these. They see their major responsibilities as raising government revenues and maintaining free and fair markets for all commodities. Direct regulation of the sale and distribution of liquor is mostly delegated to local licensing and civic authorities who perhaps are more inclined to see their roles as limiting public nuisance and overseeing a fair market rather than protecting public health. It is rare for public health considerations to be represented in the local, regional and national regulation of alcohol.

The low priority given to alcohol by governments is often reflected in the extent of national and international government funding for research, prevention and treatment. A prime example is the substantially larger budget for the US National Institute for Drug Abuse than for the US National Institute for Alcoholism and Alcohol Abuse, i.e., USD1.3 billion versus USD0.45 billion in 2020 [37,38]. This balance of effort and resourcing is also frequently reflected in national government departmental budgets and numbers of dedicated staff. For example, Health Canada’s Alcohol Policy Unit (APU) had about 1/6th of the staff positions allocated to the Office of Controlled Substances [39]. The APU has since been downsized and absorbed into another public health unit.

At various times and places, alcohol-related problems have been deemed so severe that they have become a top government priority. Total prohibitions on alcohol sales have occasionally been attempted, notably in the US and parts of Canada in the early part of the last century and, most recently, in India and South Africa as part of government measures to reduce the spread of COVID-19 (e.g., [40]). These tend to be short lived and associated with increased crime and corruption even if health and safety outcomes improve, e.g., reduced deaths from liver cirrhosis during the US Prohibition [41] and reduced violence in the South African COVID-19 lockdowns [40]. More sustainable policies to control alcohol consumption and reduce harms were introduced in the 19th and 20th centuries, principally in North America and Scandinavia, namely, direct alcohol distribution and retail systems otherwise known as “government alcohol monopolies”. Remnants of these monopolies still operate in some 17 US states, mostly restricted to sales of spirits and wine, and to some degree in all Canadian jurisdictions (even Alberta maintains government control over the distribution of alcohol) and some Scandinavian countries. The Finnish and Swedish alcohol monopolies (Alko and Systembolaget, respectively) are notable for both reporting to Ministries for Health and Social Affairs rather than following the North American practice of placing such operations within finance ministries focused on revenue collection rather than health and social outcomes [42,43]. Government alcohol control systems can facilitate ready access to the key policy levers of pricing and availability, but they also need a special mandate to do so in the interests of public health and safety rather than simple free-market economics. For this reason, the CAPE project documents both the extent of government ownership of alcohol distribution and retail systems in a jurisdiction and also to what ministry they report as an indication of the public purpose they are intended to fulfil [18].

Perhaps the most telling indication of the low priority afforded to alcohol and public health policy globally is the lack of alcohol-specific national and international laws. Room et al. [44] have made the case for an international treaty on alcohol noting such treaties exist for tobacco and illicit drug trade. In Canada, we have a federal Tobacco Act and a federal Cannabis Act but no Alcohol Act [19]. The legalisation of cannabis in Canada casts the lack of legislation and regulation of alcohol in sharp relief, especially when considering the substantially lower harms and costs of cannabis in contrast to alcohol [24].

### 3.4. Effectiveness of Alcohol Industry Influence

Globally, nationally and sub-nationally, alcohol industry groups and companies apply their immense resources to influence public debates on alcohol-related harm and policies to address this harm so as to protect their marketplace. Babor [45] described some of the ways alcohol industry-funded groups have sought to influence research agendas and shape public discussions around alcohol-related harm and prevention policies. He highlighted how these activities help to confuse public discussion of health issues and policy options and are a convenient way to demonstrate ‘corporate responsibility’ in their attempts to avoid taxation and regulation. McCambridge et al. [46] have further analysed how industry-funded social aspects bodies such as DrinkAware in the UK attempt to work closely with government agencies to influence policy agendas away from evidence-based policies that might restrict alcohol markets. Needless to say, industry groups can use their financial clout to directly influence the agendas of political parties through donations and direct lobbying in ways that are not available to the public health community.

Petticrew and colleagues published a thematic analysis of industry-sponsored publications on the role of alcohol as a risk factor for cancer. They showed how the evidence for alcohol’s causal role in cancer is downplayed, ignored or even denied with heavy emphasis on the strength of other rival risk factors [47]. There are also many examples of industry-sponsored organisations extolling various health benefits from using their product with little discussion of the growing critical literature of this interpretation (e.g., [48]). A comprehensive analysis of direct alcohol industry funding for alcohol research reported that this had increased by 56% in recent decades [49].

The CAPE project has worked to identify the extent of alcohol industry involvement in the development and implementation of alcohol policies in Canada, provincially and federally [49,50]. A key indicator used is whether alcohol industry representatives are directly involved in the development of government alcohol strategies. In 2017, it was noted that Canada’s National Alcohol Strategy Advisory Council had several Canadian alcohol manufacturers represented, a situation that has been subsequently remedied. These efforts, and many others besides, work towards creating a positive climate of opinion around alcohol and discourage effective government action to restrict its consumption.

Industry players have also shown they are not afraid to directly counter efforts to educate the public about alcohol’s role as a contributing cause of cancer. The phenomenon of “pink washing” has been widely documented whereby industry groups support cancer awareness activities [51]. A prime Canadian example some years ago was the creation of Mike’s Hard Pink Lemonade in association with the Canadian Cancer Associations breast cancer awareness week [52], i.e., an alcohol producer associating itself with a cancer prevention activity while marketing a product which contributes directly to cancer [26]. More recently, an initiative taken by the Yukon government in Canada to place cancer warning labels on all products sold in a major liquor store in its capital Whitehorse as part of a Health Canada-funded research project was shut down as a result of thinly veiled legal threats from major drinks producers [53].

### 3.5. Ineffectiveness of Public Health Advocacy

Perhaps surprisingly, even agencies concerned with the promotion of public health have often worked in opposition to or ignored the potential of evidence-based alcohol policies to improve public health. The ”Pink-Washing” of breast cancer noted above is just one example of perverse strategies from the public health field. Amin et al. [54] conducted a formal review of the websites of the Organisation for Economic Co-operation and Development cancer control agencies in 2017 as to the extent to which they identified alcohol as a risk factor for cancer. Websites in all countries except the USA and Canada at that time identified alcohol as a Class 1 carcinogen. While the US cancer agency recommended using taxation as a means of reducing tobacco-related cancers, no such action was recommended in relation to alcohol taxes. Canada is also an example of an OECD country with no national advocacy agency dedicated to promoting effective alcohol policy.

### 3.6. Summary and Ways Forward

There could not be a starker contrast between government responses to the COVID-19 pandemic versus those to the health and safety problems associated with alcohol use, despite the related harms often being similar in scale [30]. Naturally, precautions to avoid contracting a potentially deadly virus that might make you seriously ill or kill you within weeks are more likely to be supported than measures to reduce the availability and affordability of an enjoyable intoxicant, despite its short- and long-term risks. However, while the harms from COVID-19 have been counted and reported daily through multimedia throughout the course of the pandemic crisis, the majority of people in most countries are unaware of the serious harms (e.g., cancer) of alcohol consumption and of the full scale of its burden on health and well-being. Since greater than 90% of associated mortality is partially alcohol attributable, this important contribution often flies under the radar, and is not recorded on death certificates or routinely reported by health authorities. It should not be surprising, therefore, that effective government action to reduce our ability to access and purchase a favourite recreational drug, widely believed to be innocuous or even beneficial to health, is often not taken. If one adds to this the existence of powerful commercial vested interest groups who are prepared to threaten litigation against governments that exercise their legal right to warn consumers of alcohol’s health risks by placing cancer warning labels [53], it is a wonder that any government action at all is ever taken to inform citizens of alcohol’s health risks, let alone implement effective policies to reduce its consumption.

The evidence that the minority of people who understand that alcohol is a carcinogen, for example, are more likely to support evidence-based alcohol strategies (e.g., [50,55]) should be a wake-up call to the public health community to increase awareness of alcohol-related harms. While awareness rising and educational strategies in general have been decried as lacking evidence for changing population behaviours (e.g., [11]), they could likely have a critical role as a means of creating a more favourable environment for the implementation of more directly effective policies [18].

The CSUCH project in Canada has begun the difficult task of enumerating both the partially and wholly alcohol-attributable deaths and adverse health outcomes for all its jurisdictions, for all years and in comparison with other popular psychoactive substances [24]. The estimates are presented to the public and to policymakers in a variety of formats and through a variety of media, including a web-based data visualisation tool (see: https://csuch.ca (access date 11 August 2021)) for individuals to explore and download figures and data tables. Such a resource on its own is completely insufficient to substantially impact national policies on such weighty matters as taxation and pricing, but, if communicated effectively to stakeholders with an interest in, for example, preventing cancer, heart disease, violence and/or substance use disorders, it could help turn the tide of public opinion.

Perhaps the single most effective tool to raise awareness of alcohol’s potentially harmful effects would be mandated warning labels of the variety trialled by the Yukon Liquor Corporation (see Figure 1). The Health Canada-funded evaluations of this intervention demonstrated increased awareness of key messages (i.e., cancer risk, national drinking guidelines and standard drink contents) among those who recalled seeing the messages (e.g., [56]), greater intentions to reduce alcohol intake [57] and significantly reduced per capita consumption at the intervention site [58]. These labels were likely effective as they were carefully developed on the basis of published research regarding what makes a warning label noticeable, relevant and memorable (unlike the static, small, black and white US labels). A unique feature of warning labels as a means of raising awareness of alcohol-related issues is that they are more likely to be seen and recalled by those who would perhaps most benefit from the information they contain, i.e., those who consume alcohol most frequently [59].

What else might the public health community attempt by way of increasing the likelihood that effective alcohol policies might be implemented by governments? Most critically, putting pressure on their governments to give public health rather than industry groups a place at the policy-making table. At the very least, they can attempt to hold governments accountable by regularly monitoring and reporting the extent to which present policies match up with gold-standard best practices for reducing alcohol-related harm. The CAPE project, and other similar projects in Europe and the USA, draw attention to the disparities between actual implementation and best practice evidence-based policies. The CAPE project has extensively engaged with stakeholders and relevant government sectors (e.g., health, liquor regulators and government retailers, finance) to communicate clearly using publicly available and transparent indicators of effective practice [60]. This has involved going beyond lofty ideals and exhortations to “reduce alcohol’s availability” or “increase the price of alcohol” and instead reported on tangible, verifiable performance indicators such as the density of different types of liquor outlets, the extent of private versus government ownership, whether alcohol prices are keeping up with the cost of living and the level of minimum prices [18]. We have discovered that, contrary to what might be expected, across Canada’s 13 jurisdictions, examples of best practice could be found somewhere in every one of the 11 policy domains identified earlier. In fact, if identified best practices were implemented uniformly across Canada, the national score would have been 87%, i.e., an A grade. It may well be that government failures to implement effective alcohol policies are at least partly due to a lack of awareness of what is effective policy rather than a lack of willingness to implement this. However, then again, Canada may be an exception in this regard given that it still has significant, if eroding, government controls over alcohol’s distribution and retail sale(e.g., [61]).

A final thought. This essay was written in the middle of 2021 as the world is struggling to emerge from the COVID-19 pandemic. It has been widely documented that alcohol controls have weakened in many countries and that the alcohol industry has lobbied effectively to persuade governments to remove restrictions on their trade that have been in place for generations (e.g., [62]). However, there is also a significant opportunity at this moment for governments to take effective actions on alcohol to achieve a variety of both public health and fiscal outcomes. The population at large may now be more sensitised to the importance of public health measures to protect the population’s health and safety. There is also a substantial shortfall in government revenues as a consequence of economic downturns during the pandemic crisis. Alcohol pricing, taxation and availability policies could be a key part of the process of recovery. A combination of MUP and increased alcohol taxes has been shown to be a means of delivering such diverse benefits as improved public health outcomes, increased government revenues and greater industry profits [3]. Restrictions on outlet densities or at least moratoria on the issuing of new liquor licences could also help the financial sustainability of existing businesses. Such restrictions on the alcohol market could enable producers and retailers to make greater profits from selling less alcohol while also helping to shore up much-needed government revenues and reducing related harms.

## Figures and Tables

**Figure 1 nutrients-13-02846-f001:**
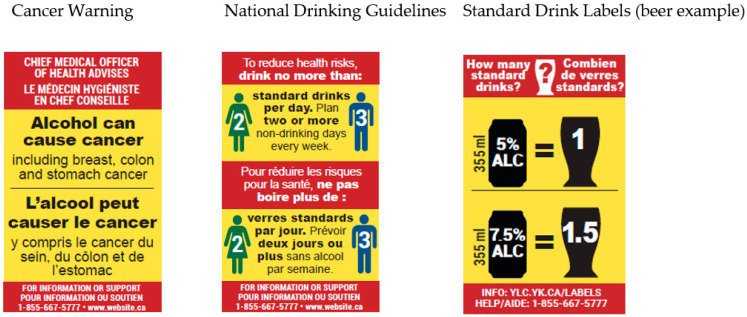
Warning labels trialled in the Yukon Territory, Canada.

**Table 1 nutrients-13-02846-t001:** CAPE provincial and territorial alcohol policy domains and weights.

Direct Policy Domains	Effectiveness (out of 5)	Scope (out of 5)	Weight ^1^ (out of 25)
**1. Pricing and Taxation**	5	5	**25**
**2. Physical Availability of Alcohol**	4	4	**16**
**3. Impaired Driving Countermeasures**	5	3	**15**
**4. Marketing and Advertising Controls**	3	5	**15**
**5. Minimum Legal Drinking Age**	4	3	**12**
**6. Screening, Brief Intervention and Referral**	3	3	**9**
**7. Liquor Law Enforcement:**	3	3	**9**
**Indirect Policy Domains**	**Facilitation** **(out of 5)**	**Scope** **(out of 5)**	**Weight ^2^** **(out of 25)**
**8. Alcohol Control System**	5	5	**25**
**9. Alcohol Strategy**	4	5	**20**
**10. Monitoring and Reporting**	4	4	**16**
**11. Health and Safety Messaging**	**3**	**4**	**12**

^1^ Weight (direct) = Effectiveness × Scope. ^2^ Weight (indirect) = Facilitation × Scope. Weights are boldfaced to emphasise these were the final estimates of potential policy impacts.

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
