# Peer review of "Government Options to Reduce the Impact of Alcohol on Human Health: Obstacles to Effective Policy Implementation"

_nutrients, 2021, doi:10.3390/nu13082846_

Round 1
Reviewer 1 Report
The paper addresses an important topic. Overall, the insights have been very good. Nether the less, some sub-chapter have been addressed a bit lengthy while other sub-chapters have been only touched shortly.
Positive arguments in regard to moderate alcohol consumption have been briefly mentioned but not taken into account further. This a pity as a literature review should clearly address both negative but also positive aspects.
All in all, I consider your pice of work as very insightful - even though only one dimensional. Nether the less, I suggest to publish it as it is.
Author Response
Many thanks for your kind comments and willingness to accept the manuscript as is. I agree that if there was more space the issue of health benefits could have been discussed in more detail. My assessment of the evidence is that there is no firm foundation for these and there are now multiple empirical and theoretical reasons to view them as the result of systematic bias and confounding - but that is a much bigger discussion!
Reviewer 2 Report
Dear Authors,
I have read with much interest and pleasure your manuscript. Only two doubts appeared. First is about the title. Would you consider "alcohol" instead of "alcoholic beverages"and what would you think about "Obstacles for effective policy implementation instead" of "Why are they under-utilised". I wonder whether you take into account the change of "Scepticism about the extent of alcohol related harm" to "Lack of general awareness of the extent of alcohol-related harm".
Yours sincerely,
Reviewer
Author Response
Thank you for your kind comments, they are greatly appreciated. I like your recommendations for wording changes and have edited the manuscript accordingly.
Reviewer 3 Report
The authors should consider a title change to better reflect the content of the manuscript and to interest and impact the correct readers. A possible better title could be something like: “Underutilized government options to reduce impact of alcoholic beverages on human health and Public Health evidenced strategies: A short review.”
The authors have submitted a very timely and interesting manuscript, which deserves consideration from other researchers but also, and more importantly, public health and government officials. They provide many examples of why some effective strategies are not implemented with evidence. They also provide several great suggestions for governments and health officials to consider moving post-COVID-19.
The manuscript is very well written and cites most of the relevant studies.
The authors should be commended on this work.
Author Response
Thank you so much for this positive assessment of our paper, it is greatly appreciated! The title of the paper has been changed to address your recommendation.